# Efficacy and safety of cAMP signalling-biased GLP-1 analogue ecnoglutide monotherapy versus placebo in patients with type 2 diabetes (EECOH-1): a multi-centre, randomised, double-blind, placebo-controlled, phase 3 trial

Ecnoglutide is a cAMP-biased GLP-1 analogue developed for the treatment of type 2 diabetes mellitus (T2DM) and obesity. We conducted a randomised, double-blind, placebo-controlled, phase 3 trial to evaluate the efficacy and safety of ecnoglutide in adults with T2DM inadequately controlled with diet and exercise alone or with a single oral hypoglycaemic agent. The primary endpoint was change in glycated haemoglobin ($HbA_{1c}$) from baseline at week 24. Between 29 December 2022 and 12 June 2024, 211 participants from 32 medical centres in China were randomised (2:2:1:1) to receive double-blind, once-weekly ecnoglutide (0.6 mg [n = 69] or 1.2 mg [n = 71]) or volume-matched placebo (0.6 mg [n = 36] or 1.2 mg [n = 35]) for 24 weeks. The randomisation, stratified by baseline $HbA_{1c}$ (≤8.5% or >8.5%), was conducted via an interactive web response system. Ecnoglutide and placebo were identical in appearance to achieve masking. The trial was completed. All randomised participants received ≥1 dose of the assigned treatment and thus were included for analyses. At week 24, the least squares mean changes from baseline in $HbA_{1c}$ were −1.96% (95% CI −2.18 to −1.73) with ecnoglutide 0.6 mg and −2.43% (95% CI −2.65 to −2.20) with ecnoglutide 1.2 mg versus −0.87% (−1.09 to −0.65) with placebo. The estimated treatment differences versus placebo were −1.09% (95% CI −1.40 to −0.77; p = 0.0003) with ecnoglutide 0.6 mg and −1.56% (95% CI −1.87 to −1.24; p < 0.0001) with ecnoglutide 1.2 mg. Ecnoglutide represents a potential monotherapy option for T2DM. This trial was registered at clinicaltrials.gov with the registration number NCT05680155.

Type 2 diabetes mellitus (T2DM), a progressive metabolic disease primarily characterised by abnormal glucose metabolism, poses an enormous burden on individuals as well as health systems across the world[1–3]. The goal of T2DM management is to reduce the risk of associated complications through optimal glycaemic control. Despite a wide range of available treatment options, a large proportion of patients still cannot achieve glycated haemoglobin ($HbA_{1c}$) treatment targets[4,5]. Furthermore, glycaemic management should consider minimising undesired effects such as hypoglycaemia and bodyweight gain[6], which has proven to be challenging with traditional glucose-lowering medications.

✉e-mail: zhudalong@nju.edu.cn; shaohui.bing@sciwindbio.com

The advert of single glucagon-like peptide-1 (GLP-1) receptor agonists such as semaglutide and dual glucose-dependent insulino-tropic polypeptide (GIP) and GLP-1 receptor agonist tirzepatide has transformed the treatment landscape of T2DM. They can control gly-caemia effectively without inducing severe hypoglycaemia or body-weight gain[7–9]. Apart from glycaemic control, GLP-1 receptor agonists provide other clinical benefits, including bodyweight loss, cardiovas-cular risk reduction, and improvement in renal outcomes among others[10–12]. Therefore, they are an effective treatment option for T2DM and have been recommended by various guidelines[13–15].

Ecnoglutide, also known as XW003, is a potent cyclic adenosine monophosphate (cAMP)-biased GLP-1 analogue, containing an alanine-to-valine substitution at position 8 as well as an 18-C fatty acid con-jugation at the lysine 30 side chain[16]. cAMP bias is hypothesised to enhance the clinical efficacy of GLP-1 receptor agonists through reducing internalisation of the GLP-1 receptor and enhancing insulin secretion[17]. In a preclinical study, ecnoglutide showed a stronger binding affinity towards the GLP-1 receptor and more potent efficacy in reducing blood glucose and bodyweight than semaglutide, an unbiased GLP-1 receptor agonist[16]. In a phase 1 trial among healthy volunteers, once-weekly injections of ecnoglutide exhibited favourable safety and tolerability profiles and a half-life ranging from 124 to 138 h, indicating its potential as a long-acting regimen[16]. In a phase 2 trial among individuals with T2DM, once-weekly injections of ecnoglutide at doses of 0.4, 0.8, and 1.2 mg resulted in more pronounced improvements versus placebo in glycaemic control and bodyweight, supporting its potential as a treatment option for T2DM[18]. In the phase 3 SLIMMER trial among individuals with overweight or obesity, once-weekly injections of ecnoglutide at 1.2, 1.8, and 2.4 mg showed superior and sustained reduction in bodyweight versus placebo, supporting its potential use for weight management[19].

Here we report the findings from a phase 3 trial, EECOH-1, which investigated the efficacy and safety of once-weekly injections of ecnoglutide at doses of 0.6 mg and 1.2 mg versus placebo in adults with T2DM inadequately controlled with diet and exercise alone or with a single oral hypoglycaemic agent. In this work, we show that ecnoglutide monotherapy significantly improved glycaemic control versus placebo and was well-tolerated in this population.

## Results

### Study participants

Between 29 December 2022 and 12 June 2024, 300 participants were assessed for eligibility, and 211 of them were randomly assigned to receive 0.6 mg ecnoglutide ($n = 69$), 1.2 mg ecnoglutide ($n = 71$), pla-cebo volume-matched to 0.6 mg ecnoglutide ($n = 36$), or placebo volume-matched to 1.2 mg ecnoglutide ($n = 35$) (Fig. 1) for 24 weeks in a double-blind manner. All randomised participants received ≥1 dose of the assigned treatment, thereby comprising the full analysis set (FAS) and safety set, and 9 (4%) of them discontinued treatment prematurely in the double-blind period, including 3 from the 0.6 mg ecnoglutide group, 4 from the 1.2 mg ecnoglutide group, and 2 from the pooled placebo group (referred to as the placebo group hereafter). Following the double-blind period, out of the 211 randomised participants, 195 (92.4%) entered the 28-week open-label period and received treatment with ecnoglutide (0.6 or 1.2 mg) and 5 (2.6%) of them discontinued treatment prematurely, including 2 from the 0.6 mg ecnoglutide group and 3 from the 1.2 mg ecnoglutide group. Across the two peri-ods, the reasons for treatment discontinuation included physicians' decisions and participants' own requests among others. Time to study drug discontinuation is shown in Kaplan-Meier plots in Supplemen-tary Fig. 1.

During the double-blind treatment period, 3 participants in the placebo group received rescue therapy because of hyperglycaemia,

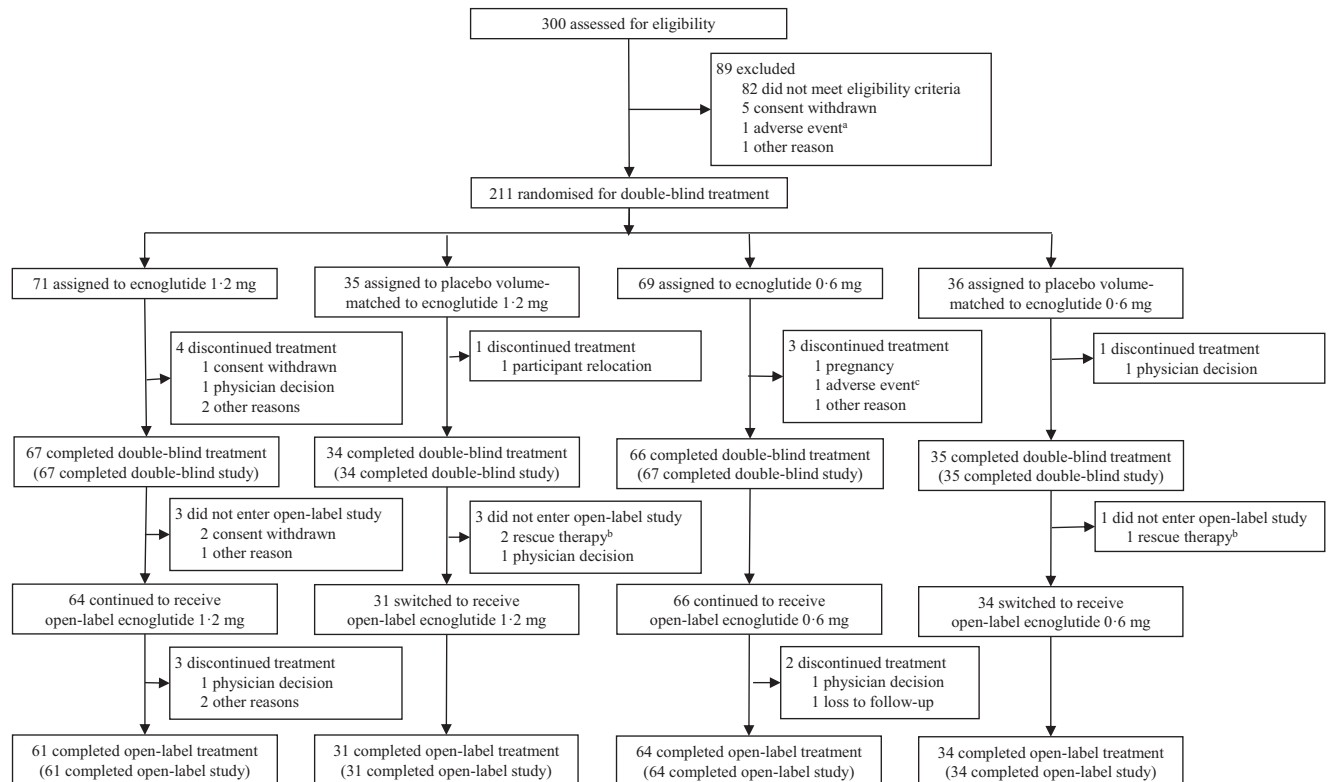

**Fig. 1 | Trial profile.** Notes: [a]One participant who had a pre-dose adverse event during the run-in period and was not randomised. [b]Three participants who did not enter the open-label study after 24 weeks of the double-blind treatment due to rescue therapy were considered as completing the trial according to the protocol. [c]One participant discontinued treatment because of decreased appetite.

**Table 1 | Baseline characteristics (full analysis set)**

| Characteristic | Ecnoglutide 1.2 mg (N = 71) | Ecnoglutide 0.6 mg (N = 69) | Placebo (N = 71) | Total (N = 211) |
|---|---|---|---|---|
| Age (years) | 52.5 (11.5) | 52.1 (10.8) | 51.4 (10.4) | 52.0 (10.9) |
| **Sex** | | | | |
| Female | 28 (39%) | 30 (43%) | 26 (37%) | 84 (40%) |
| Male | 43 (61%) | 39 (57%) | 45 (63%) | 127 (60%) |
| Bodyweight (kg) | 71.53 (12.05) | 74.25 (15.20) | 73.69 (11.31) | 73.14 (12.93) |
| Height (cm) | 165.30 (8.03) | 164.83 (9.80) | 165.22 (9.34) | 165.12 (9.04) |
| BMI (kg/m$^2$) | 26.35 (3.31) | 27.23 (3.65) | 27.23 (3.20) | 26.93 (3.40) |
| Waist circumference (cm) | 93.26 (8.94) | 94.02 (10.55) | 94.60 (8.36) | 93.96 (9.29) |
| Diabetes duration (years) | 3.66 (3.99) | 4.10 (4.20) | 3.00 (3.02) | 3.58 (3.78) |
| Prior antihyperglycemic medication use | 35 (49) | 41 (59) | 26 (37) | 102 (48) |
| **HbA$_{1c}$** | | | | |
| In % | 8.51 (0.83) | 8.54 (0.80) | 8.51 (0.81) | 8.52 (0.81) |
| In mmol/mol | 69.51 (9.10) | 69.84(8.72) | 69.51 (8.89) | 69.62 (8.86) |
| ≤8.5% | 38 (54%) | 36 (52%) | 37 (52%) | 111 (53%) |
| >8.5% | 33 (46%) | 33 (48%) | 34 (48%) | 100 (47%) |
| Fasting plasma glucose (mmol/L) | 9.67 (1.86) | 9.67 (1.76) | 9.81 (1.82) | 9.72 (1.81) |
| eGFR$_{MDRD}$ (ml/min/1.73m$^2$) | 124.7 (35.1) | 121.9 (29.9) | 113.5 (25.9) | 120.0 (30.7) |

Notes: Data are mean (SD) or n (%). Data are for all randomised participants. $N$ = all randomly assigned participants who took at least one dose of the study drug. *BMI*, body mass index; *eGFR$_{MDRD}$*, estimated glomerular filtration rate based on the Modification of Diet in Renal Disease equation; *HbA$_{1c}$*, glycated haemoglobin.

and none in the ecnoglutide groups required rescue therapy. During the open-label treatment period, 3 participants, with one each from the two ecnoglutide groups and the 0.6 mg placebo group, required rescue therapy because of hyperglycaemia.

Baseline characteristics were similar across the three groups (Table 1). At baseline, the mean (standard deviation, SD) age was 52.0 (10.9) years, 127 (60.2%) were male, with an average (SD) of 8.52 (0.81) for HbA$_{1c}$, 26.93 kg/m$^2$ (3.40) for body mass index (BMI), and 3.58 years (3.78) for T2DM duration.

### Primary outcome

The primary endpoint was met, and both doses of ecnoglutide significantly reduced HbA$_{1c}$ more than placebo for both primary and secondary efficacy estimands. At week 24, the least squares mean (LSM) change (95% confidence interval [CI]) from baseline in HbA$_{1c}$ as per the treatment policy estimand was −1.96% (−2.18 to −1.73) with 0.6 mg ecnoglutide, −2.43% (−2.65 to −2.20) with 1.2 mg ecnoglutide, and −0.87% (−1.09 to −0.65) with placebo (Fig. 2B, Table 2). The estimated treatment difference (ETD) versus placebo was −1.09% (95% CI −1.40 to −0.77; $p$ = 0.0003) with 0.6 mg ecnoglutide and −1.56% (95% CI −1.87 to −1.24; $p$ < 0.0001) with 1.2 mg ecnoglutide (Fig. 2B, Table 2). Significantly greater decreases from baseline in HbA$_{1c}$ with both 0.6 mg and 1.2 mg ecnoglutide versus placebo were evident since the first assessment at week 4 (Fig. 2A). The cumulative distribution of change from baseline in HbA$_{1c}$ at week 24 is presented in Supplementary Fig. 2. At week 24, the LSM change (95% CI) from baseline in HbA$_{1c}$ as per the hypothetical estimand was −1.96% (−2.18 to −1.73) with 0.6 mg ecnoglutide, −2.43% (−2.65 to −2.20) with 1.2 mg ecnoglutide, and −0.85% (−1.07 to −0.62) with placebo. The ETD versus placebo was −1.11% (95% CI −1.43 to −0.79; $p$ = 0.0002) with 0.6 mg ecnoglutide and −1.58% (95% CI −1.90 to −1.26; $p$ < 0.0001) with 1.2 mg ecnoglutide.

Prespecified sensitivity analyses all supported the conclusions of the primary analysis for the primary efficacy endpoint, showing similar and significant ETDs with both 0.6 and 1.2 mg ecnoglutide versus placebo (Supplementary Fig. 3). Subgroup analysis also showed that both 0.6 and 1.2 mg ecnoglutide consistently reduced HbA$_{1c}$ versus placebo to a larger extent across subgroups stratified by sex, age, baseline HbA$_{1c}$, BMI, and hypoglycaemia agent use prior to screening (Supplementary Fig. 4).

### Secondary outcomes

At week 24, the proportion of participants who achieved HbA$_{1c}$ < 7.0% was 68.1% (47/69) with 0.6 mg ecnoglutide and 80.3% (57/71) with 1.2 mg ecnoglutide, both significantly higher than 21.1% (15/71) with placebo ($p$ < 0.0001 for both); the proportion of participants who achieved HbA$_{1c}$ ≤ 6.5% was 52.2% (36/69) with 0.6 mg ecnoglutide and 76.1% (54/71) with 1.2 mg ecnoglutide, both significantly higher than 12.7% (9/71) with placebo ($p$ < 0.0001 for both); the proportion of participants who achieved normoglycaemia (HbA$_{1c}$ ≤ 5.7%) was 10.1% (7/69) with 0.6 mg ecnoglutide and 35.2% (25/71) with 1.2 mg ecnoglutide, compared to 0% (0/71) with placebo (Supplementary Table 1). Similarly, more participants achieved the composite endpoint of HbA$_{1c}$ < 7.0% without severe hypoglycaemia and without bodyweight gain in both the 0.6 mg and 1.2 mg ecnoglutide groups than in the placebo group (58.0% [40/69] and 71.8% [51/71] versus 18.3% [13/71], $p$ < 0.0001 for both) (Fig. 2C, Supplementary Table 1).

At week 24, the LSM change (95% CI) from baseline in fasting plasma glucose (FPG) was −2.89 mmol/L (−3.24 to −2.53) with 0.6 mg ecnoglutide, −3.32 mmol/L (−3.68 to −2.97) with 1.2 mg ecnoglutide, and −1.21 mmol/L (−1.56 to −0.86) with placebo (Table 2). The ETD versus placebo was −1.68 mmol/L (95% CI −2.17 to −1.18; $p$ < 0.0001) with 0.6 mg ecnoglutide and −2.11 mmol/L (95% CI −2.61 to −1.62; $p$ < 0.0001) with 1.2 mg ecnoglutide (Fig. 2D, Table 2). The LSM change (95% CI) from baseline in 2h-postprandial plasma glucose (PPG) was −5.95 mmol/L (−6.69 to −5.21) with 0.6 mg ecnoglutide, −7.42 mmol/L (−8.17 to −6.67) with 1.2 mg ecnoglutide, and −1.81 mmol/L (−2.54 to −1.08) with placebo. The ETD versus placebo was −4.14 mmol/L (95% CI −5.18 to −3.10; $p$ < 0.0001) with 0.6 mg ecnoglutide and −5.61 mmol/L (95% CI −6.66 to −4.56; $p$ < 0.0001) with 1.2 mg ecnoglutide (Table 2). Significantly larger reductions in mean seven-point self-monitored blood glucose (SMBG) and mean postprandial glucose excursion were also observed with 0.6 mg and 1.2 mg ecnoglutide versus placebo ($p$ < 0.001 for all; Fig. 2E, Table 2).

At week 24, the LSM change (95% CI) from baseline in bodyweight was −3.04 kg (−3.68 to −2.40) with 0.6 mg ecnoglutide, −3.21 kg (−3.84 to −2.58) with 1.2 mg ecnoglutide, and −1.45 kg (−2.08 to −0.82) with placebo. The ETD versus placebo was −1.59 kg (95% CI −2.48 to −0.69; $p$ = 0.0005) with 0.6 mg ecnoglutide and −1.76 kg (95% CI −2.65 to −0.87; $p$ = 0.0001) with 1.2 mg ecnoglutide (Table 2). The cumulative

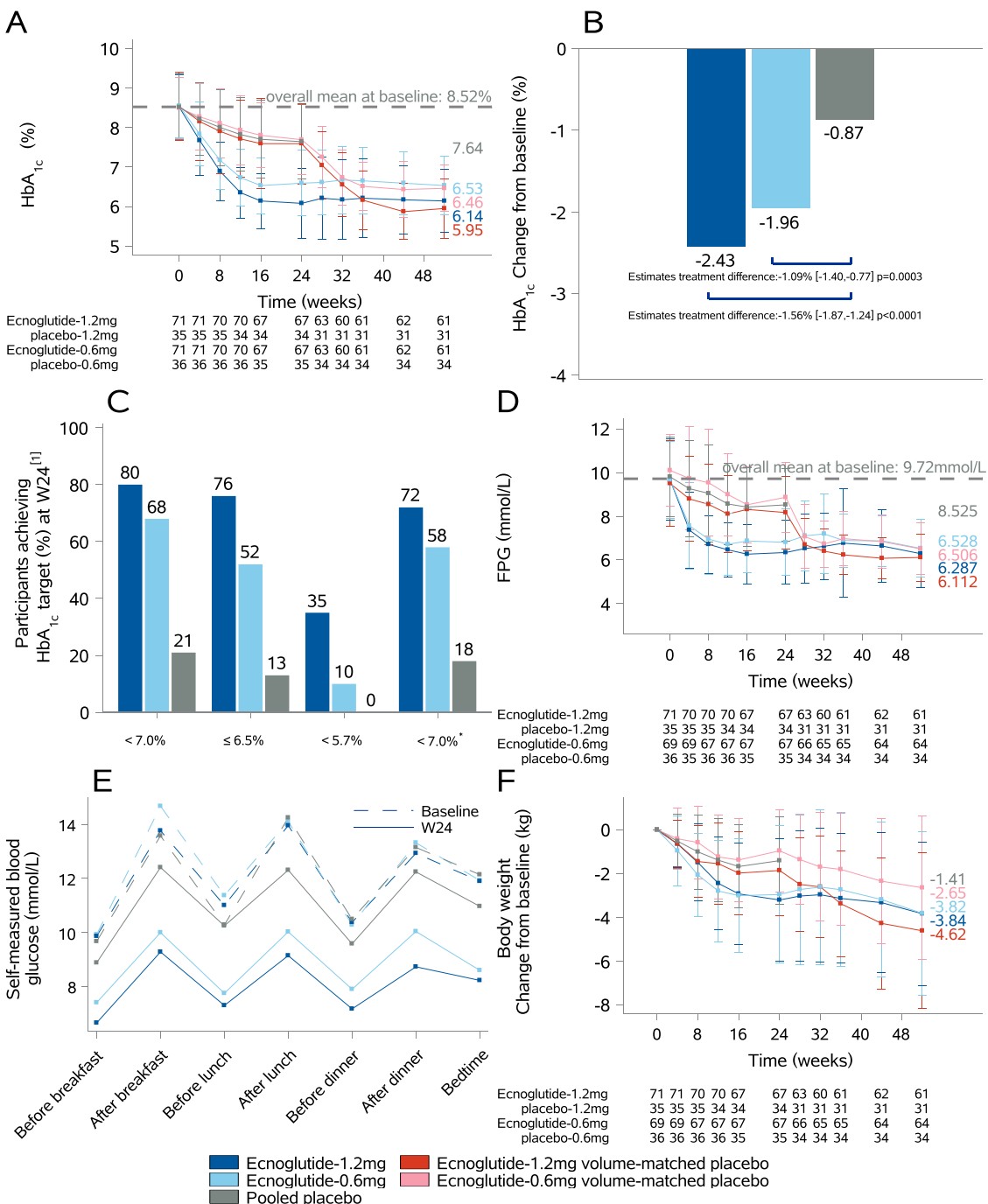

**Fig. 2 | Efficacy outcomes of 0.6 mg and 1.2 mg ecnoglutide once weekly versus placebo at weeks 24 and 52.** Data are mean (SD), unless otherwise noted, and participants in the placebo groups switched to ecnoglutide after week 24. **A** Change from baseline in HbA$_{1c}$ over time. **B** Change from baseline in HbA$_{1c}$ at week 24 from MMRM analysis (estimated treatment differences are LSM [95% CI] at week 24 [full analysis set] and *p*-values were obtained from two-sided *t*-tests with multiplicity adjusted). **C** Proportion of participants reaching HbA$_{1c}$ targets (<7.0%, ≤6.5%, and

<5.7%) and the composite endpoint (HbA$_{1c}$ < 7.0%, no severe hypoglycaemia, and no bodyweight gain) analysis at week 24. **D** Change from baseline in FPG over time. **E** Seven-point SMBG profiles at baseline and week 24. **F** Change from baseline in bodyweight over time. Source data are provided as a Source Data file. CI = confidence interval. FPG = fasting plasma glucose. HbA$_{1c}$ = glycated haemoglobin. LSM = least squares mean. MMRM = mixed model for repeated measures. SD = standard deviation. SMBG = self-monitored blood glucose.

distribution of change from baseline in bodyweight at week 24 is presented in Supplementary Fig. 5. The LSM percentage change (95% CI) from baseline in bodyweight was −4.51% (−5.43 to −3.58) with 0.6 mg ecnoglutide, −4.74% (−5.65 to −3.82) with 1.2 mg ecnoglutide, and −2.02% (−2.92 to −1.11) with placebo. The ETD versus placebo was −2.49% (95% CI −3.79 to −1.20; *p* = 0.0002) with 0.6 mg ecnoglutide and −2.72% (95% CI −4.01 to −1.43; *p* < 0.0001) with 1.2 mg ecnoglutide (Table 2).

At week 24, a bodyweight reduction of ≥5% from baseline was achieved by significantly more participants in the 0.6 mg and 1.2 mg ecnoglutide groups than in the placebo group (39.1% [27/69] and 43.7% [31/71] versus 11.3% [8/71], *p* = 0.0002 and *p* < 0.0001, respectively); more participants achieved a ≥10% bodyweight reduction in the 0.6 mg and 1.2 mg ecnoglutide groups than in the placebo group, although the differences were not statistically significant (Supplementary Table 1).

**Table 2 | Efficacy measures at week 24 (full analysis set)**

| | Ecnoglutide 1.2 mg (N = 71) | | | Ecnoglutide 0.6 mg (N = 69) | | | Placebo (N = 71) |
|---|---|---|---|---|---|---|---|
| | LSM change from baseline (95% CI) | ETD vs placebo (95% CI) | p value | LSM change from baseline (95% CI) | ETD vs placebo (95% CI) | p value | LSM change from baseline (95% CI) |
| HbA$_{1c}$ (%) | −2.43 (−2.65 to −2.20) | −1.56 (−1.87 to −1.24) | <0.0001 | −1.96 (−2.18 to −1.73) | −1.09 (−1.40 to −0.77) | 0.0003 | −0.87 (−1.09 to −0.65) |
| HbA$_{1c}$ (mmol/mol) | −50.0 (−52.44 to −47.59) | −17.01 (−20.43 to −13.57) | <0.0001 | −44.89 (−47.34 to −42.43) | −11.88 (−15.32 to −8.44) | 0.0003 | −33.01 (−35.42 to −30.59) |
| Fasting plasma glucose (mmol/L) | −3.32 (−3.68 to −2.97) | −2.11 (−2.61 to −1.62) | <0.0001 | −2.89 (−3.24 to −2.53) | −1.68 (−2.17 to −1.18) | <0.0001 | −1.21 (−1.56 to −0.86) |
| 2h-postprandial plasma glucose (mmol/L) | −7.42 (−8.17 to −6.67) | −5.61 (−6.66 to −4.56) | <0.0001 | −5.95 (−6.69 to −5.21) | −4.14 (−5.18 to −3.10) | <0.0001 | −1.81 (−2.54 to −1.08) |
| **7-Point SMBG (mmol/L)** | | | | | | | |
| Mean Glucose | −3.99 (−4.43 to −3.54) | −2.81 (−3.45 to −2.17) | <0.0001 | −3.36 (−3.81 to −2.91) | −2.18 (−2.82 to −1.55) | <0.0001 | −1.18 (−1.63 to −0.72) |
| Postprandial glucose excursion | −1.58 (−1.91 to −1.24) | −1.03 (−1.50 to −0.55) | <0.0001 | −1.38 (−1.72 to −1.04) | −0.83 (−1.31 to −0.35) | 0.0008 | −0.55 (−0.89 to −0.21) |
| Bodyweight (kg) | −3.21 (−3.84 to −2.58) | −1.76 (−2.65 to −0.87) | 0.0001 | −3.04 (−3.68 to −2.40) | −1.59 (−2.48 to −0.69) | 0.0005 | −1.45 (−2.08 to −0.82) |
| Bodyweight (%) | −4.74 (−5.65 to −3.82) | −2.72 (−4.01 to −1.43) | <0.0001 | −4.51 (−5.43 to −3.58) | −2.49 (−3.79 to −1.20) | 0.0002 | −2.02 (−2.92 to −1.11) |
| Waist circumference (cm) | −3.35 (−4.13 to −2.57) | −2.15 (−3.25 to −1.06) | 0.0001 | −2.75 (−3.54 to −1.97) | −1.56 (−2.66 to −0.46) | 0.0058 | −1.19 (−1.97 to −0.42) |
| Hip circumference (cm) | −2.96 (−3.55 to −2.37) | −1.34 (−2.17 to −0.50) | 0.0018 | −2.49 (−3.09 to −1.89) | −0.87 (−1.71 to −0.03) | 0.0425 | −1.62 (−2.21 to −1.03) |

Notes: P-values for the primary endpoint were obtained by two-sided t-tests based on MMRM, with multiplicity adjusted, and P-values for the secondary endpoints here were obtained by two-sided t-tests based on MMRM, without multiplicity adjusted. CI, confidence interval; ETD, estimated treatment difference; HbA$_{1c}$ glycated haemoglobin; LSM, least square mean; MMRM, mixed models for repeated measures; SMBG, self-monitoring of blood glucose.

At week 24, waist circumference decreased from baseline by an LSM of 2.75 cm with 0.6 mg ecnoglutide, 3.35 cm with 1.2 mg ecnoglutide, and 1.19 cm with placebo. The ETDs versus placebo were significant for both ecnoglutide groups (Table 2). Hip circumference also significantly decreased from baseline with both doses of ecnoglutide versus placebo (Table 2).

At week 24, homoeostasis model assessments of β-cell function (HOMA-β) significantly increased with both doses of ecnoglutide than with placebo. Homoeostasis model assessments of insulin resistance (HOMA-IR) decreased more with ecnoglutide than with placebo, though the differences did not reach statistical significance. No significant changes compared to placebo were observed for fasting insulin levels for either dose of ecnoglutide (Supplementary Table 1). At week 24, decreases from baseline in low-density lipoprotein (LDL) cholesterol were observed in the ecnoglutide groups, and these changes were not significantly different from those with placebo; a significant decrease from baseline in triglycerides versus placebo was observed with 1.2 mg ecnoglutide versus placebo, not with 0.6 mg ecnoglutide (Supplementary Table 1).

Efficacy data at week 52 are summarised in Supplementary Table 2. In the two groups who received ecnoglutide throughout, improvements in glycaemia control and bodyweight in the double-blind treatment period were maintained during the open-label treatment period (Fig. 2). In each placebo group, improvements in glycaemia control and bodyweight, among other efficacy measures were observed after switching to ecnoglutide (Fig. 2, Supplementary Table 2). At week 52, mean (SD) changes from baseline in HbA$_{1c}$ were −2.01% (0.934) in the 0.6 mg ecnoglutide group, −2.05% (0.849) in the placebo-to-0.6-mg-ecnoglutide group, −2.34% (1.120) in the 1.2 mg ecnoglutide group, and −2.45% (0.947) in the placebo-to-1.2-mg-ecnoglutide group (Supplementary Table 2), and mean (SD) changes from baseline in bodyweight were −3.82 (3.754) kg in the 0.6 mg ecnoglutide group, −2.65 (3.280) kg in the placebo-to-0.6-mg-ecnoglutide group, −3.84 (3.281) kg in the 1.2 mg ecnoglutide group, and −4.62 (3.555) kg in the placebo-to-1.2-mg-ecnoglutide group (Supplementary Table 2).

### Safety outcomes

During the double-blinded treatment period, treatment-emergent adverse events (TEAEs) occurred in 78.3% (54/69) of the 0.6 mg ecnoglutide group, 77.5% (55/71) of the 1.2 mg ecnoglutide group, and 63.4% (45/71) of the placebo group (Table 3). Serious TEAEs were reported in 2.9% (2/69) of the 0.6 mg ecnoglutide group, 4.2% (3/71) of the 1.2 mg ecnoglutide group, and 5.6% (4/71) of the placebo group. TEAEs led to premature treatment discontinuation in 1 (1.4%) participant in each treatment group (Table 3). No deaths occurred in any group. The most frequently reported TEAEs by system organ class were gastrointestinal, metabolic, and nutritional disorders. The most commonly reported gastrointestinal events for ecnoglutide were diarrhoea and nausea (Table 3). Other TEAEs with a >5% incidence included decreased appetite, increased lipase, upper respiratory tract infection, urinary tract infection, and asthenia, among others. Most adverse events in the ecnoglutide groups were mild to moderate in severity, and the incidence was the highest during the dose-escalation period and decreased over time (Supplementary Fig. 6).

Hypoglycaemia (identified via scheduled laboratory visit) was reported in 2 (2.9%) participants in the 0.6 mg ecnoglutide group, 5 (5.6%) in the 1.2 mg ecnoglutide group, and 1 (1.4%) in the placebo group (Table 3). No severe hypoglycaemia, pancreatitis, or gallbladder-related disorders were reported with ecnoglutide in this study. Elevated lipase and amylase levels were reported by more participants in the ecnoglutide groups (11.6% [8/69] and 4.3% [3/69] in the 0.6 mg group and 12.7% [9/71] and 2.8% [2/71] in the 1.2 mg group) than in the placebo group (2.8% [2/71] and 0 [0/71]). These elevations did not appear to be dose-related (Supplementary Fig. 7), and were not

**Table 3 | Treatment-emergent adverse events (TEAEs) during the double-blind treatment period (safety set[a])**

| Participants, n (%) | Ecnoglutide 1.2 mg (N = 71) | Ecnoglutide 0.6 mg (N = 69) | Placebo (N = 71) |
|---|---|---|---|
| Any TEAEs | 55 (77.5) | 54 (78.3) | 45 (63.4) |
| Treatment-related | 44 (62.0) | 40 (58.0) | 20 (28.2) |
| Grade ≥3 TEAEs | 5 (7.0) | 3 (4.3) | 5 (7.0) |
| Treatment-related | 3 (4.2) | 1 (1.4) | 0 |
| Serious TEAEs | 3 (4.2) | 2 (2.9) | 4 (5.6) |
| Treatment-related | 1 (1.4) | 0 | 0 |
| TEAEs leading to treatment discontinuation | 1 (1.4) | 1 (1.4) | 1 (1.4) |
| TEAEs leading to death | 0 | 0 | 0 |
| **TEAEs occurring in ≥5% of participants in any treatment group (preferred term)** | | | |
| Decreased appetite | 19 (26.8) | 15 (21.7) | 3 (4.2) |
| Asthenia | 10 (14.1) | 1 (1.4) | 2 (2.8) |
| Lipase increased | 9 (12.7) | 8 (11.6) | 2 (2.8) |
| Nausea | 9 (12.7) | 5 (7.2) | 7 (9.9) |
| Diarrhoea | 8 (11.3) | 17 (24.6) | 4 (5.6) |
| Abdominal distension | 6 (8.5) | 3 (4.3) | 1 (1.4) |
| Dizziness | 5 (7.0) | 2 (2.9) | 0 |
| Upper respiratory tract infection | 5 (7.0) | 7 (10.1) | 3 (4.2) |
| Sinus tachycardia | 4 (5.6) | 1 (1.4) | 1 (1.4) |
| Hyperlipidaemia | 2 (2.8) | 3 (4.3) | 5 (7.0) |
| Urine leucocyte positive | 2 (2.8) | 2 (2.9) | 5 (7.0) |
| Anaemia | 1 (1.4) | 4 (5.8) | 2 (2.8) |
| Flatulence | 1 (1.4) | 0 | 4 (5.6) |
| Urinary tract infection | 1 (1.4) | 6 (8.7) | 7 (9.9) |
| **TEAEs of special interest occurring in ≥5% of participants in any treatment group (preferred term)** | | | |
| Decreased appetite | 19 (26.8) | 15 (21.7) | 3 (4.2) |
| Nausea | 9 (12.7) | 5 (7.2) | 7 (9.9) |
| Diarrhoea | 8 (11.3) | 17 (24.6) | 4 (5.6) |
| Abdominal distension | 6 (8.5) | 3 (4.3) | 1 (1.4) |
| Hypoglycaemia (blood glucose <3.9 mmol/L) | 4 (5.6) | 2 (2.9) | 1 (1.4) |
| Flatulence | 1 (1.4) | 0 | 4 (5.6) |

Note: [a] The safety set comprised all participants who received ≥1 dose of study treatment and safety evaluation after treatment initiation.

associated with clinical symptoms or signs, pancreatitis, or hepatic disorders.

Pulse rate increased by a mean of 3.0 beats/min with ecnoglutide 0.6 mg and 5.4 beats/min with 1.2 mg ecnoglutide versus a mean decrease of 1.6 beats/min with placebo at week 24. Mean systolic blood pressure decreased by 4.4 mmHg with 0.6 mg ecnoglutide and a mean of 5.2 mmHg with 1.2 mg ecnoglutide versus 2.1 mmHg with placebo. Changes in diastolic blood pressure from baseline were similar across the three groups at week 24 (Supplementary Table 3).

During the open-label period, the proportions of participants reporting TEAEs in the two ecnoglutide groups were lower than their counterparts in the double-blind period, and the incidences of TEAEs in the two groups switching from placebo to ecnoglutide were similar to what was reported by the two ecnoglutide groups in the double-blind period (Supplementary Table 4). Across the four treatment groups, gastrointestinal disorders were still the most frequently reported events, which were mostly transient and mild to moderate in severity. No pancreatitis or severe hypoglycaemic events were reported.

## Discussion

In this phase 3 trial, compared with placebo, once-weekly injections of ecnoglutide at doses of 1.2 mg and 0.6 mg demonstrated significant improvement in glycaemic control in participants with T2DM inadequately controlled with diet and exercise alone or with a single oral hypoglycaemic agent. Both doses of ecnoglutide were superior to placebo in reducing $HbA_{1c}$, without increasing the risk of hypoglycaemia. Reductions in $HbA_{1c}$ were observed by the first study assessment at week 4 and sustained until the end of treatment at week 52. The magnitude of the reductions in $HbA_{1c}$ observed for ecnoglutide was dose-dependent. At week 24, up to 80.3% in the 1.2 mg ecnoglutide group and 68.1% in the 0.6 mg ecnoglutide group versus 21.1% in the placebo group reached the $HbA_{1c}$ target of <7.0% as recommended by the American Diabetes Association (ADA)[20], and up to 35.2% of participants in the 1.2 mg ecnoglutide group reached normoglycaemia ($HbA_{1c}$ < 5.7%). More importantly, the proportion of participants who achieved the composite endpoint of $HbA_{1c}$ < 7.0% without severe glycaemia and without bodyweight gain was also significantly higher in the ecnoglutide groups than in the placebo group at week 24 (71.8% with 1.2 mg and 58.0% with 0.6 mg versus 18.3% with placebo). The beneficial effect on glycaemic control was further supported by the significant improvements in FPG, 2h-PPG, 7-Point SMBG profiles, and the β-cell function.

These results agree with the findings in the phase 2 trial of ecnoglutide in participants with T2DM that was uncontrolled with diet, exercise, or single oral glucose-lowering medication[18]. In this phase 2 study, the $HbA_{1c}$ target of <7.0% was achieved in 84% with ecnoglutide 1.2 mg and 68% with ecnoglutide 0.8 mg versus 21% with placebo at week 20. Although indirect cross-trial comparisons should be viewed with caution due to differences in trial designs, populations, and analysis methods, ecnoglutide showed comparable or greater $HbA_{1c}$ reductions versus other GLP-1-based therapies, including selective GLP-1 receptor agonists dulaglutide and supaglutide. The decreases from baseline in $HbA_{1c}$ were 1.96% and 2.43% with ecnoglutide (0.6 and 1.2 mg at week 24) in the current study, compared to 1.25% and 1.46% for dulaglutide (0.75 and 1.5 mg at week 26) in the Chinese subgroup of an East Asian trial (AWARD-CHN1), and 1.73% and 2.15% for supaglutide (1 and 3 mg at week 24) in a phase 3 trial among Chinese adults (NCT04994288)[21,22]. A head-to-head preclinical study has shown that at the same dose level, ecnoglutide significantly reduced blood glucose and $HbA_{1c}$ more than semaglutide[16], which has been shown to be more effacious in reducing $HbA_{1c}$ than dulaglutide in the phase 3 SUSTAIN 7 trial[23].

In addition to glycaemic effects, significant and sustained bodyweight reductions were observed over 52 weeks of treatment with ecnoglutide. The average bodyweight loss was 3.04 and 3.21 kg for ecnoglutide 0.6 and 1.2 mg at week 24 in the current study, numerically higher compared to 1.0 and 1.5 kg for dulaglutide (0.75 and 1.5 mg at week 26) in the Chinese subgroup of the AWARD-CHN1 trial, and 0.69 and 2.25 kg for supaglutide (1 and 3 mg at week 24) in the phase 3 trial among Chinese adults[21,22]. More importantly, at week 24, 39.1–43.7% in the ecnoglutide groups versus 11.3% in the placebo group achieved a bodyweight loss of ≥5%, meeting the ADA-recommended target range for weight loss[24]. Favourable changes in waist and hip circumferences with ecnoglutide versus placebo were also observed at week 24, providing further evidence for the beneficial effects of ecnoglutide in bodyweight management. In addition, favourable changes in systolic blood pressure, lipids, and insulin sensitivity were observed with ecnoglutide, suggesting potential cardiovascular benefits.

The observed safety profile of ecnoglutide in this study population was consistent with the known class effects of GLP-1 receptor agonists, with transient, mild to moderate gastrointestinal events as the most frequently reported TEAEs. Gastrointestinal adverse events mainly occurred in the dose escalation period and diminished over time. No new safety signals were detected in this study. No cases of

severe hypoglycaemia were reported throughout the study. Slight increases in pulse rate were noted for ecnoglutide in this study, which were consistent with observations for other GLP-1 receptor agonists[8,9,25]. Both doses of ecnoglutide were well-tolerated, which was evident in the very low rate of treatment discontinuation. Only 1.4% of participants in the two groups discontinued ecnoglutide due to TEAEs.

This study has several strengths, including the randomised, placebo-controlled design with the use of a volume-matched placebo for masking within each dose and a switch-over design to allow for minimised exposure to placebo treatment and long-term efficacy evaluation. Additionally, a relatively high proportion of participants completed the study, giving fairly complete data. This study also has limitations. Firstly, it was conducted in China only, so the observed findings may not be fully translatable to other ethnic/racial populations. However, since GLP-1 receptor agonists have been shown to be efficacious in both Asian and non-Asian populations[26], similar efficacy and safety of ecnoglutide are expected in non-Asian populations as well. Secondly, although the trial duration was sufficiently long to assess the primary and secondary endpoints, the long-term impact of ecnoglutide on cardiovascular and renal outcomes requires longer studies to assess fully.

This study observed a good placebo effect in efficacy measures such as changes from baseline in HbA$_{1c}$, FPG, and bodyweight, similar to the placebo effects observed in another reported phase 3 trial conducted in China[27]. This placebo effect might be due to the relatively short diabetic history (mean duration, 3 years) and the high proportion (63%) of treatment-naïve participants, which may contribute to more responsiveness to non-pharmacological interventions. This effect could also be due to randomisation, participants' adherence to treatment, as well as guidance from their healthcare professionals, and participants' continued lifestyle modifications during the trial. Another phase 3 study (EECOH-2) has been conducted to assess the long-term efficacy and safety of ecnoglutide as an add-on to metformin in patients with T2DM (NCT05680129)[28]. EECOH-2 demonstrated that once-weekly ecnoglutide 0.6 mg and 1.2 mg as add-on therapy were non-inferior to dulaglutide 1.5 mg in reducing HbA$_{1c}$ in adults with T2DM inadequately controlled by metformin monotherapy[28].

In conclusion, ecnoglutide, a cAMP signalling-biased GLP-1 receptor agonist, administered at doses of 0.6 and 1.2 mg once weekly as monotherapy for T2DM, significantly improved glycaemic control, with up to 35.2% of participants achieving normoglycaemia, and robustly reduced bodyweight versus placebo, without increasing the risk of severe hypoglycaemia. Ecnoglutide was well-tolerated, with a very low rate of treatment discontinuation and a safety profile consistent with other GLP-1 receptor agonists. The favourable efficacy and safety profiles indicate that ecnoglutide is a promising initial treatment option for T2DM early in the course of the disease.

## Methods

### Study design
We conducted a randomised, double-blind, placebo-controlled, phase 3 trial across 32 sites in China. This trial was conducted in compliance with the Declaration of Helsinki, Good Clinical Practice Guidelines, and all applicable local laws and regulations. This study protocol was approved by institutional review boards or ethics committees of all participating sites, with the full list provided within **Supplementary Information** (p 2), and written informed consent was obtained from all participants prior to study entry. The study was reported according to the CONSORT 2025 statement (Supplementary Note 1).

The study comprised a 2-week screening period, a 4-week run-in period, a 24-week double-blind core treatment period, a 28-week open-label extended treatment period, and a 5-week safety follow-up (Supplementary Fig. 8).

Participants were randomly assigned (2:2:1:1) to receive once-weekly subcutaneous injections of ecnoglutide (0.6 or 1.2 mg) or volume-matched placebo (0.6 or 1.2 mg), with stratification according to baseline HbA$_{1c}$ (≤8.5% or >8.5%), via an interactive web response system. All investigators, participants, and the sponsor remained blinded to treatment assignment. Placebo and active drugs were provided in injector pens, identical in appearance.

After a 2-week screening period and a 4-week run-in period, participants were re-assessed for eligibility. Eligible participants received once weekly subcutaneous injections of study drugs in a double-blind manner for 24 weeks. At the end of 24 weeks, participants were unblinded one by one and all switched to open-label treatment. They either continued to receive ecnoglutide at the respective maintenance doses or started receiving ecnoglutide for 28 weeks, followed by a 5-week safety follow-up period.

Study drug was administered once weekly via subcutaneous injection using injector pens with a pre-set volume, following a slow dose-escalation regimen, starting at 0.3 mg (equivalent to 0.15 mL injection) with fixed double-dose increments every 4 weeks until the allocated maintenance dose was reached. The maintenance doses of 0.6 and 1.2 mg were reached at 4 and 8 weeks, respectively, in the respective ecnoglutide groups. Participants who switched from placebo to ecnoglutide during the open-label treatment period underwent the same dose-escalation procedure. If intolerable symptoms or events occurred and persisted, a lower tolerated dose could be used at the investigator's discretion. More details on the dosing regimen are available in the **Supplementary Information** (pp 2–3). Initiation of new antihyperglycemic medications, with metformin as the first choice, was only allowed for rescue therapy for persistent hyperglycaemia on the basis of prespecified criteria (**Supplementary Information**, pp 3–4).

### Participants
Key inclusion criteria included adults aged 18–75 years (inclusive) with T2DM inadequately controlled with diet and exercise alone or with only one oral hypoglycaemic agent. Eligible participants had a BMI of 20.0–35.0 kg/m² (inclusive), an HbA$_{1c}$ level of 7.5%–11.0% (inclusive) at screening, and an HbA$_{1c}$ level of 7.0%–10.5% (inclusive) at randomisation, and a FPG level of ≤13.9 mmol/L at both screening and randomisation. Participants were excluded if they had type 1 diabetes mellitus or other diabetes mellitus, received any GLP-1 drug or dipeptidyl peptidase-4 inhibitor within the preceding 3 months or insulin within the preceding 6 months (except for the ≤14-day use of insulin for co-morbidities), or experienced diabetic ketoacidosis, hyperosmolar hyperglycaemic state, lactic acidosis in diabetes, or severe chronic complications of diabetes within the prior 6 months. The full list of eligibility criteria is provided in the **Supplementary Information** (pp 4–5).

### Study endpoints and assessments
The primary efficacy endpoint was the change from baseline in HbA$_{1c}$ at week 24, assessed by the central laboratory. The secondary efficacy endpoints included proportions of participants who achieved an HbA$_{1c}$ level of <7.0% and ≤6.5% at weeks 24 and 52; proportion of participants who achieved a composite endpoint of HbA$_{1c}$ < 7.0%, no severe hypoglycaemia, and no bodyweight gain at week 24; changes from baseline at weeks 24 and 52 in FPG, 2h-PPG, seven-point SMBG profiles, fasting insulin, HOMA-β and HOMA-IR, blood lipids, bodyweight, waist circumference, and hip circumference.

Safety endpoints included the incidences of TEAEs, serious TEAEs, and TEAEs of special interest (hypoglycaemia, cardiovascular events, gastrointestinal events, pancreatitis, and gallbladder-related disorders). Other safety measurements included vital signs, physical examinations, 12-lead electrocardiograms (ECG), and laboratory

assessments. The other secondary endpoints included pharmacokinetics and immunogenicity, which will be reported separately.

## Statistical analyses

The study was designed to establish superiority for each dose of ecnoglutide versus placebo for the primary endpoint at week 24. The sample size calculation assumed at least a $-1.2\%$ difference in mean change from baseline in $HbA_{1c}$ at week 24 between ecnoglutide groups and the pooled placebo group, a common SD of 1.1%, and a drop-out rate of 20%. A sample size of 210 participants provided at least 90% power to establish superiority for an ecnoglutide dose compared with placebo (superiority margin of 0.5%) at a one-sided significance level of 0.025. Full details of the type I error control strategy for the primary endpoint are provided in the **Supplementary Information** (pp 5–6). All reported p-values for secondary endpoints are nominal because they were not adjusted for multiplicity.

Efficacy analyses were performed in the FAS, comprising all randomised participants who received ≥1 dose of study treatment. Safety analyses were conducted in the safety set, comprising all participants who received ≥1 dose of study treatment and safety evaluation after treatment initiation.

The primary efficacy endpoint was evaluated using two estimands. For the primary efficacy estimand, the treatment policy strategy was used to assess the treatment effect between ecnoglutide and placebo among all randomised participants regardless of intercurrent events (early treatment discontinuation and rescue therapy). For the secondary efficacy estimand, the hypothetical strategy was used to assess the treatment effect between ecnoglutide and placebo among all participants without intercurrent events, i.e., data collected after intercurrent events were excluded from analysis.

For the primary endpoint analysis, a mixed model for repeated measures (MMRM) was used, in which the change from baseline in $HbA_{1c}$ was used as the dependent variable, categorical baseline $HbA_{1c}$ level, visit time points, treatment grouping, and treatment by visit interaction were used as the explanatory variables. The model was used to calculate covariate-adjusted mean change from baseline in $HbA_{1c}$ of each group at week 24 and its standard error, its 95% confidence interval (CI), as well as inter-group mean difference between each ecnoglutide group and the placebo group and its 95% CI. For missing $HbA_{1c}$ values at week 24 in participants who experienced intercurrent events, imputation was performed using the retrieved dropout-based multiple imputation, or the missing at random approach if the former was not appropriate.

Prespecified sensitivity analyses were conducted to assess the robustness of results for the primary endpoint by the use of alternative data selections and methods for handling missing data, including analysis of covariance (ANCOVA), which considered baseline $HbA_{1c}$ and treatment group as exploratory variables, and the jump to reference (J2R) assumption, in which missing data was imputed based on the reference group data. Subgroup analysis was also conducted for the primary endpoint.

For the analysis of the secondary efficacy endpoints, change from baseline in FPG at week 24, and change and percentage change from baseline in bodyweight at week 24 were analysed by strategies and methods consistent with those for the primary endpoint. The proportions of participants who achieved prespecified targets of bodyweight loss or $HbA_{1c}$ reductions were analysed and compared using logistic regression, and the 95% CIs for these proportions were calculated using the Clopper-Pearson method, with missing data imputed using the non-response imputation method. Results for the other secondary efficacy endpoints were summarised descriptively without data imputation. Safety endpoints were summarised descriptively by the use of data for all randomised participants who received at least one dose of study drug (safety set). Statistical analyses were performed using SAS version 9.4. This study was registered with both ClinicalTrials.gov (NCT05680155) and China Drug Trials Registry (www.chinadrugtrials.org.cn, CTR20223156).

## Reporting summary

Further information on research design is available in the Nature Portfolio Reporting Summary linked to this article.

## Data availability

Source data are provided as a Source Data file. The full dataset and protocol are not publicly available due to data privacy laws and contractual obligations. Sciwind Biosciences will provide de-identified individual participant data underlying the reported results upon request. Data are available after acceptance of this article, with no expiration of data requests currently set. Requests should be made by contacting the corresponding authors, D. Zhu (zhudalong@nju.edu.cn) or S. Bing (shaohui.bing@sciwindbio.com), and will be evaluated within 6 months of receipt. Access will be provided for researchers after the proposed use of the data in a methodologically sound proposal for academic purposes has been approved by a review committee and receipt of a signed data access agreement with Sciwind Biosciences. Source data are provided with this paper.

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

## Acknowledgements

We thank the participants, the investigators, and their teams who took part in this study. This study was funded by Hangzhou Sciwind Biosciences Co., Ltd. The study sponsor was involved in study design and protocol development and was responsible for data collection, data analysis, data interpretation, and writing of the report. The decision to submit the report for publication was made by all authors, who had full access to the data. The authors would like to acknowledge Minguang Zhang (YOUR-DATA, Beijing, China) for statistical consultation and Jinlong Guo, PhD (Costello Medical, Singapore) for medical writing support in compliance with Good Publication Practice (2022). Both services were funded by Hangzhou Sciwind Biosciences Co., Ltd. Part of the data from this study were presented at the American Diabetes Association's 84th Scientific Sessions, 21–24 June 2024, in Orland, Floria, the United States of America.

## Author contributions

D.Z., W.W., H.P., Y.L., S.X., S.B., Q.Z., J.N., L.Y., W.G., L.G., X.L., Y.B., M.G., G.T., J.M., B.W., X.Z., B.S., S.P., K.W., X.S., X.Z., L.F., Y.L., Y.Lu, D.H., C.J., T.P., H.X., J.H., and H.D. designed and performed the study. D.Z., H.P., M.Y., S.X., S.B., Q.Z., and F.J. analysed and interpreted the data. D.Z., S.B., and S.X. draughted the manuscript. All authors approved the final submitted paper.

## Competing interests

D. Z., W. W., G. T., J. M., B. W., X. Z., B. S., S. P., K. W., X. S., X. Z., L. F., Y. L., Y. Lu, D. H., C. J., T. P., H. X., J. H., H. D. received funding from Sciwind Biosciences to their institutions as trial investigators. S. B., F. J., Q. Z., M. Y., L. G., X. L., Y. B., M. G., J. N., L. Y., W. G., Y. L., H. P. are employees of Hangzhou Sciwind Biosciences. S X. is an employee of Sciwind Biosciences.

## Additional information

Dalong Zhu [1,2] ✉, Weimin Wang[1], Guoyu Tong[1], Jianhua Ma[3], Binhong Wen[4], Xin Zheng[5], Bimin Shi[6], Shuguang Pang[7], Kun Wang[8], Xiaoxia Shi[9], Xianghua Zhang[10], Liujun Fu[11], Yang Liu[12], Yibing Lu[13], Debin Huang[14], Chengxia Jiang[15], Tianrong Pan [16], Haibo Xue[17], Jie Han[18], Hongcheng Ding[19], Shaohui Bing [20] ✉, Feifei Jiang[20], Qing Zheng [20], Ming Yang[20], Lei Guan[20], Xingquan Liu[20], Jing Ning[20], Yue Bu[20], Mengying Guo[20], Liu Yang[20], Wanjun Guo[20], Yao Li[20], Susan Xu[21] & Hai Pan[20]

[1]Department of Endocrinology, Endocrine and Metabolic Disease Center, Nanjing Drum Tower Hospital, The Affiliated Hospital of Nanjing University Medical School, Nanjing, Jiangsu, China. [2]Branch of National Clinical Research Center for Metabolic Diseases, Nanjing, Jiangsu, China. [3]Department of Endocrinology, Nanjing First Hospital, Nanjing Medical University, Nanjing, Jiangsu, China. [4]Endocrine and metabolic disease Diagnosis and Treatment Center, The People's Hospital of Liaoning Province, Shenyang, Liaoning, China. [5]Department of Endocrinology, Beijing Boai Hospital, Beijing, Beijing, China. [6]Department of Endocrinology, The First Affiliated Hospital of Soochow University, Suzhou, Jiangsu, China. [7]Department of Endocrinology, Central Hospital Affiliated to Shandong First Medical University, Jinan, Shandong, China. [8]Department of Endocrinology, Nanjing Jiangning Hospital, Nanjing, Jiangsu, China. [9]Endocrinology Department, The First Affiliated Hospital of Nanyang Medical College, Nanyang, Henan, China. [10]General Medical Department, Yueyang Central Hospital, Yueyang, Hunan, China. [11]Endocrinology and Metabolism Clinic, The First Affiliated Hospital of Henan University of Science & Technology, Luoyang, Henan, China. [12]National Metabolic Management Center, Daqing People's Hospital, Daqing, Heilongjiang, China. [13]Department of Endocrinology, The Second Affiliated Hospital of Nanjing Medical University, Nanjing, Jiangsu, China. [14]Endocrine Metabolism Department, The Third Hospital of Changsha, Changsha, Hunan, China. [15]Department of Endocrinology, Yibin Second People's Hospital, Yibin, Sichuan, China. [16]Endocrinology Department, The Second Hospital of Anhui Medical University, Hefei, Anhui, China. [17]Department of Endocrinology and Metabolism, Binzhou Medical University Hospital, Binzhou, Shandong, China. [18]Endocrinology Department, Hebei Petro China Central Hospital, Langfang, Hebei, China. [19]Department of Endocrinology, Renmin Hospital, Hubei University of Medicine, Shiyan, Hubei, China. [20]Hangzhou Sciwind Biosciences, 400 Fucheng Rd, Rm 901 9F Bldg 2, Qiantang District, Hangzhou, Zhejiang, China. [21]Sciwind Biosciences, 3001 Bishop Drive, Suite 300, San Ramon, CA, USA. ✉e-mail: zhudalong@nju.edu.cn; shaohui.-bing@sciwindbio.com

