## [Transparent Peer Review file · Nature Communications]

Efficacy and safety of cAMP signalling-biased GLP-1 analogue ecnoglutide monotherapy versus placebo in patients with type 2 diabetes (EECOH-1): a multi-centre, randomised, double-blind, placebo-controlled, phase 3 trial

Corresponding Author: Mr shaohui Bing

Version 1:

Reviewer comments:

Reviewer #1

(Remarks to the Author)

This manuscript reports the results of a multicenter, randomized, double-blind, placebo-controlled phase 3 trial (EECOH-1) evaluating the efficacy and safety of ecnoglutide, a cAMP-biased GLP-1 receptor agonist, in patients with type 2 diabetes. While ecnoglutide improved glycemic control and reduced body weight with a safety profile similar to existing GLP-1 receptor agonists, the proposed role of biased signaling remains speculative. The study does not provide mechanistic or translational evidence to demonstrate that the cAMP-biased profile translates into clinical benefit.

In the absence of a clear mechanistic link or comparative advantage over established GLP-1RAs, the novelty and impact of the work may be insufficient for Nature Communications. Given the limited novelty for a general audience, the manuscript may be more suitable for a specialty journal dedicated to clinical diabetes research.

Specific comments :

To meaningfully assess the clinical value of biased agonism, future studies should directly compare ecnoglutide with existing GLP-1RAs in terms of efficacy, tolerability, durability, or β -cell function.

The manuscript places repeated emphasis on the concept that ecnoglutide is a “cAMP-biased GLP-1 receptor agonist.” However, the clinical relevance of this signaling bias remains unclear. Given the current lack of mechanistic or biomarker data in human subjects, it would be more appropriate to present the potential role of cAMP-biased signaling as a hypothesis rather than a demonstrated mechanism.

This placebo-controlled design does not allow assessment of whether the cAMP-biased profile of ecnoglutide offers clinical advantages over existing GLP-1RAs such as semaglutide. While the efficacy and safety results appear acceptable, the added value of biased signaling remains uncertain. The reviewer suggests that the authors clarify in the discussion that no conclusions can be drawn regarding superiority over established agents, and consider briefly comparing the results with historical data on currently approved GLP-1RAs for context.

Reviewer #2

(Remarks to the Author)

Reviewer #3

(Remarks to the Author)

The manuscript NCOMMS-25-24131A-Z reports on a randomised, double-blind, placebo-controlled phase 3 trial, which

investigated the efficacy and safety of once-weekly ecnoglutide monotherapy at doses of 0.6 mg and 1.2 mg versus placebo in adults with T2DM inadequately controlled with diet and exercise alone or with a single oral hypoglycaemic agent across 32 sites in China. Ecnoglutide is a novel cyclic adenosine monophosphate (cAMP) signalling-biased GLP-1 analogue developed for the treatment of type 2 diabetes mellitus and obesity.

The primary endpoint was met, and both doses of ecnoglutide significantly reduced HbA1c more than placebo for both primary and secondary efficacy estimands. The proportion of participants who achieved the composite endpoint of HbA1c<7.0% without severe glycaemia and without bodyweight gain was also significantly higher in the ecnoglutide groups as compared to the placebo group at week 24. In the two groups that received ecnoglutide throughout, improvements in glucose control and bodyweight achieved in the double-blind treatment period were maintained during the open-label treatment period. The observed safety profile of ecnoglutide in this study population was consistent with the known class effects of GLP-1 receptor agonists, with transient, mild to moderate gastrointestinal events as the most frequently reported TEAEs.

Thus, in the trial reported, ecnoglutide has demonstrated superiority over placebo in Chinese participants with type 2 diabetes. The manuscript presents clinically relevant data adding information for a new promising treatment option within the group of GLP-1 based therapies for people with type 2 diabetes early in the course of the disease. The methodology used is sound and allows for the conclusions to be made.

Here are some critical points to be considered:

1. 211 participants were randomised (2:2:1:1) to receive ecnoglutide 1.2 mg (n=71), 0.6 mg (n=69), or volume-matched placebo (n=71) for 24 weeks, and thereafter 195 entered the open-label period and received ecnoglutide (0.6 or 1.2 mg) for 28 weeks. Data on change in HbA1c, FPG and body weight for the 28-week open-label period for the patients from the placebo groups are not presented and should be added to Figure 2 A,D,F.
2. One of the inclusion criteria was use of only one oral hypoglycemic agent within the previous 12 weeks, but no further use within 2 weeks prior to screening. The percentage of patients that were treatment-naïve and those with use of an oral hypoglycemic agent within the previous 12 weeks till 2 weeks prior to screening should be mentioned in the text. Probably the study was not powered to analyse and provide separate data on the 2 groups - already treated with a single oral hypoglycaemic agent and treatment-naïve, but at least some data need to be provided in the text.
3. More Grade ≥ 3 TEAEs have been reported in the placebo groups as compared to the ecnoglutide treated groups which needs to be clarified. The authors should discuss this in the text. How was Hyperlipidaemia defined as an adverse effect?
4. Hypoglycemia, cardiovascular effects, gastrointestinal events, pancreatitis and gallbladder related disorders, have been considered adverse events of special interest (AESIs), but just data on hypoglycaemia is presented on Table 3. Data on the other AESIs should also be added in Table 3.
5. One of the main limitations of the study is that it was conducted in China and therefore the findings are limited to the Chinese population. This has been pointed out in the manuscript. As the data refer to a Chinese population, comparison with data for other GLP-1 RA and dual GLP-/GIP RA conducted in the general population is not appropriate.
6. The structure of the manuscript needs to be rearranged as the Methods section follows the Discussion section and the conclusions of the study.
7. Supplementary tables and figures have not been provided for review.
8. The most recent Standards of care of the American Diabetes Association Professional Practice Committee (2025) should be referred to in the manuscript.

Reviewer #4

(Remarks to the Author)

This manuscript reports a phase 3 randomized controlled trial evaluating ecnoglutide, a cAMP-biased GLP-1 receptor agonist, in Chinese patients with type 2 diabetes. The study demonstrated significant improvements in HbA1c, body weight, and other metabolic markers over 24 weeks. The design is appropriate and the findings are promising. However, several key points require clarification to strengthen the manuscript.

Major Comments :

1. The rationale for selecting only 0.6 mg and 1.2 mg doses is unclear. Previous trials with ecnoglutide in obesity used up to 2.4 mg (10.1016/S2213-8587(25)00141-X). Please explain why higher doses were not tested in this diabetes population. Was it due to tolerability, expected efficacy plateau, or phase 2 data?
2. The 1.2 mg group lost an average of 3.2 kg. Please briefly compare this with other GLP-1RAs and discuss its relevance for metabolic comorbidities.
3. Although the study includes a 28-week open-label extension following the 24-week double-blind period, the manuscript provides minimal information regarding its design and outcomes. While a brief mention of week 52 data is made in the Results section with reference to Supplementary Table 2 and Figure 2, the purpose of the open-label phase (e.g., assessment of long-term efficacy or safety) is not clearly defined in the Methods, and the week 52 data are not summarized in the main text. Please clarify the objectives, key endpoints, and analysis plan for the open-label period, and consider including a concise summary of week 52 findings in the Results section.
4. HbA1c reduction in the placebo group was -0.87% , which is higher than expected. Please discuss possible reasons, such as lifestyle changes, enhanced monitoring, or statistical regression. Fasting glucose and body weight also improved in the placebo group. Please consider whether this influenced effect size estimation and whether sensitivity analyses were performed.
5. The 1.2 mg group had a mean increase of 5.4 bpm in heart rate. Although this is a known class effect, it may have implications for patients with underlying cardiovascular conditions. Additional data on QTc, nocturnal HR, or cardiovascular events would be helpful.

Version 2:

Reviewer comments:

Reviewer #1

(Remarks to the Author)

There are no further comments.

Reviewer #2

(Remarks to the Author)

Reviewer #3

(Remarks to the Author)

When reviewing the manuscript I made 8 main comments that the authors should address.

In the revised version of the manuscript which has been resubmitted to Nature Communications, the authors have considered all the comments and remarks except for point 6, but I agree with them for not accepting this comment as it is not in accordance with the journal requirements.

I have no further comments to add.

Reviewer #4

(Remarks to the Author)

The authors responded appropriately to all the raised issues and comments.No further comments.

Comments from Reviewer #1

This manuscript reports the results of a multicenter, randomized, double-blind, placebo-controlled phase 3 trial (EECOH-1) evaluating the efficacy and safety of ecnoglutide, a cAMP-biased GLP-1 receptor agonist, in patients with type 2 diabetes. While ecnoglutide improved glycemic control and reduced body weight with a safety profile similar to existing GLP-1 receptor agonists, the proposed role of biased signaling remains speculative. The study does not provide mechanistic or translational evidence to demonstrate that the cAMP-biased profile translates into clinical benefit.

In the absence of a clear mechanistic link or comparative advantage over established GLP-1RAs, the novelty and impact of the work may be insufficient for Nature Communications. Given the limited novelty for a general audience, the manuscript may be more suitable for a specialty journal dedicated to clinical diabetes research.

Specific comments:

To meaningfully assess the clinical value of biased agonism, future studies should directly compare ecnoglutide with existing GLP-1RAs in terms of efficacy, tolerability, durability, or β -cell function.

The manuscript places repeated emphasis on the concept that ecnoglutide is a "cAMP-biased GLP-1 receptor agonist." However, the clinical relevance of this signaling bias remains unclear. Given the current lack of mechanistic or biomarker data in human subjects, it would be more appropriate to present the potential role of cAMP-biased signaling as a hypothesis rather than a demonstrated mechanism.

This placebo-controlled design does not allow assessment of whether the cAMP-biased profile of ecnoglutide offers clinical advantages over existing GLP-1RAs such as semaglutide. While the efficacy and safety results appear acceptable, the added value of biased signaling remains uncertain. The reviewer suggests that the authors clarify in the discussion that no conclusions can be drawn regarding superiority over established agents, and consider briefly comparing the results with historical data on currently approved GLP-1RAs for context.

Response: Thank you for your feedback. First, we would like to clarify that, according to the *Technical Guidelines for Clinical Development of Drugs for Type 2 Diabetes in Adults* issued in 2023 by the Center for Drug Evaluation (CDE), National Medical Products Administration (NMPA), China,¹ to meet exposure and benefit-risk assessment requirements, pivotal clinical trials for non-insulin therapies in type 2 diabetes typically require at least two studies, usually: (1) a confirmatory trial evaluating the safety and efficacy of the drug as monotherapy, and (2) a confirmatory trial evaluating the benefits and risks of the drug in combination with metformin; placebo and an active control should be used as a comparator for the monotherapy study and the combination study, respectively.¹ Consistent with this recommendation, we have conducted two phase 3 clinical trials (EECOH-1 and EECO-2) for the use of ecnoglutide in patients with type 2 diabetes. EECO-1, which is what the current manuscript reports, evaluated the efficacy and safety of ecnoglutide monotherapy versus placebo in patients with type 2 diabetes inadequately controlled with diet and exercise alone or with a single oral hypoglycaemic agent. EECO-2 evaluated the efficacy and safety of ecnoglutide versus dulaglutide (another GLP-1 analogue) as add-on therapy to metformin in patients with type 2 diabetes inadequately controlled with metformin monotherapy. Therefore, we did assess the efficacy and safety of ecnoglutide versus another GLP-1 analogue.

The findings from EECOH-2 have been published recently.² At the time of study conduct, the limited availability of semaglutide in China restricted the feasibility of using it as a comparator and the GLP-1 receptor agonist dulaglutide was a standard of care and readily available in China at that time, making it a suitable and pragmatic choice. EECOH-2 reported that once-weekly ecnoglutide 0.6 mg and 1.2 mg were non-inferior and superior to dulaglutide 1.5 mg in reducing HbA_{1c} in adults with type 2 diabetes and elevated glucose concentrations on metformin monotherapy and both doses of ecnoglutide were well tolerated.² Within EECOH-2, the efficacy of ecnoglutide in glycaemic control was further supported by significant improvements versus dulaglutide in fasting plasma glucose, as well as in both pre- and post-prandial blood glucose readings from 7-point self-monitoring of blood glucose. Furthermore, fewer participants in each ecnoglutide group required rescue therapy for hypoglycaemia during the study. Apart from glycaemic effects, both doses of ecnoglutide induced statistically significantly greater reductions in bodyweight, waist circumference, hip circumference, and triglycerides than dulaglutide 1.5 mg. Therefore, EECOH-2 indeed demonstrated the cAMP-biased GLP-1 receptor agonist ecnoglutide provides additional clinical benefits beyond the non-biased GLP-1 receptor agonist dulaglutide.

Additionally, in the phase 3 trial (SLIMMER) of ecnoglutide in adults with overweight or obesity,³ ecnoglutide induced robust and dose-dependent weight loss and appeared to show comparable or higher efficacy compared with other GLP-1-based therapies based on cross-trial comparison. For example, the 15.1% placebo-adjusted bodyweight reduction with 2.4 mg ecnoglutide at week 48 is numerically higher than 8.5% reported for semaglutide 2.4 mg at week 44 in STEP 7⁴ and aligns with tirzepatide 15 mg outcomes (15.1% at week 52) in SURMOUNT-CN.⁵ Therefore, this SLIMMER study also supports the potential benefits with the cAMP-biased GLP-1 receptor agonist ecnoglutide versus a non-biased GLP-1 receptor agonist.

Collectively speaking, both EECOH-2 and SLIMMER support that the added clinical advantages from the cAMP-biased agonism versus the unbiased agonism exist, although studies comparing the clinical effects of a biased GLP-1 analogue with those of a pharmacokinetically matched but balanced GLP-1 analogue would be warranted to fully understand the added value of this innovative mechanism.^{6,7} Additionally, we would like to highlight that a head-to-head preclinical study has showed that at the same dose level, ecnoglutide reduced blood glucose and HbA_{1c} significantly more than semaglutide.⁸

We also would like to clarify that the comparisons with historical data of semaglutide, tirzepatide, and dulaglutide in the same treatment setting had been previously included in the manuscript. Based on Review 2's Comment 5, it is not appropriate to compare this study with other studies conducted in the general population (non-China population). Therefore, we have kept the comparison with the findings from the Chinese subgroup of the AWARD-CHN1 trial of dulaglutide conducted in the same treatment setting (Pages 14 to 15, Lines 328 to 336),⁹ and removed the comparisons with historical data of semaglutide and tirzepatide because both drugs do not have phase 3 data from a Chinese population in the same treatment setting. Additionally, we have added a comparison with historical phase 3 data of another GLP-1 analogue supaglutide (recently approved in China) among Chinese patients from the same treatment setting (Pages 14 to 15, Lines 328 to 336).

Comments from Reviewer #2

Response: Thank you for the explanation.

Comments from Reviewer #3

The manuscript NCOMMS-25-24131A-Z reports on a randomised, double-blind, placebo-controlled phase 3 trial, which investigated the efficacy and safety of once-weekly ecnoglutide monotherapy at doses of 0.6 mg and 1.2 mg versus placebo in adults with T2DM inadequately controlled with diet and exercise alone or with a single oral hypoglycaemic agent across 32 sites in China. Ecnoglutide is a novel cyclic adenosine monophosphate (cAMP) signalling-biased GLP-1 analogue developed for the treatment of type 2 diabetes mellitus and obesity.

The primary endpoint was met, and both doses of ecnoglutide significantly reduced HbA1c more than placebo for both primary and secondary efficacy estimands. The proportion of participants who achieved the composite endpoint of HbA1c<7.0% without severe glycaemia and without bodyweight gain was also significantly higher in the ecnoglutide groups as compared to the placebo group at week 24. In the two groups that received ecnoglutide throughout, improvements in glucose control and bodyweight achieved in the double-blind treatment period were maintained during the open-label treatment period. The observed safety profile of ecnoglutide in this study population was consistent with the known class effects of GLP-1 receptor agonists, with transient, mild to moderate gastrointestinal events as the most frequently reported TEAEs. Thus, in the trial reported, ecnoglutide has demonstrated superiority over placebo in Chinese participants with type 2 diabetes. The manuscript presents clinically relevant data adding information for a new promising treatment option within the group of GLP-1 based therapies for people with type 2 diabetes early in the course of the disease. The methodology used is sound and allows for the conclusions to be made.

Response: Thank you for the summary. We have provided responses to your comments below.

Here are some critical points to be considered:

1. 211 participants were randomised (2:2:1:1) to receive ecnoglutide 1.2 mg (n =71), 0.6 mg (n=69), or volume-matched placebo (n=71) for 24 weeks, and thereafter 195 entered the open-label period and received ecnoglutide (0.6 or 1.2 mg) for 28 weeks. Data on change in HbA1c, FPG and body weight for the 28-week open-label period for the patients from the placebo groups are not presented and should be added to Figure 2 A,D,F.

Response: Thank you for the suggestion. We have added data after week 24 for the two placebo groups to Parts A, D, F of Figure 2.

2. One of the inclusion criteria was use of only one oral hypoglycemic agent within the previous 12 weeks, but no further use within 2 weeks prior to screening. The percentage of patients that were treatment-naïve and those with use of an oral hypoglycemic agent within the previous 12 weeks till

as an abnormal laboratory finding can also be considered an adverse event based on investigator assessment.

4. Hypoglycemia, cardiovascular effects, gastrointestinal events, pancreatitis and gallbladder related disorders, have been considered adverse events of special interest (AESIs), but just data on hypoglycaemia is presented on Table 3. Data on the other AESIs should also be added in Table 3.

Response: Thank you for the suggestion. We have updated **Table 3** and added the most common AESIs ($\geq 5\%$ in any treatment group by preferred term) to it.

5. One of the main limitations of the study is that it was conducted in China and therefore the findings are limited to the Chinese population. This has been pointed out in the manuscript. As the data refer to a Chinese population, comparison with data for other GLP-1 RA and dual GLP-/GIP RA conducted in the general population is not appropriate.

Response: Thank you for the suggestion. We have removed the comparisons with data for other GLP-1-based therapies in non-China populations and retained (and added) the comparisons with data for other approved GLP-1-based therapies from Chinese patients in the same treatment setting (Pages 14 to 15, Lines 328 to 336).

6. The structure of the manuscript needs to be rearranged as the Methods section follows the Discussion section and the conclusions of the study.

Response: Thank you for the suggestion. According to the journal requirements, the Methods section should be placed after the Discussion section, so we have not adjusted the manuscript structure as you suggested.

7. Supplementary tables and figures have not been provided for review.

Response: Thank you for pointing out this. The supplementary tables and figures are now attached for your review.

8. The most recent Standards of care of the American Diabetes Association Professional Practice Committee (2025) should be referred to in the manuscript.

Response: Thank you for the suggestion. The latest ADA guidance is referred to in the manuscript now (Page 4, Lines 82 to 84; Page 14, Lines 313 to 317; Page 15, Lines 347 to 349).

Comments from Reviewer #4

This manuscript reports a phase 3 randomized controlled trial evaluating ecnoglutide, a cAMP-biased GLP-1 receptor agonist, in Chinese patients with type 2 diabetes. The study demonstrated significant improvements in HbA1c, body weight, and other metabolic markers over 24 weeks. The design is appropriate and the findings are promising. However, several key points require clarification to strengthen the manuscript.

Major Comments:

1. The rationale for selecting only 0.6 mg and 1.2 mg doses is unclear. Previous trials with

ecnoglutide in obesity used up to 2.4 mg (10.1016/S2213-8587(25)00141-X). Please explain why higher doses were not tested in this diabetes population. Was it due to tolerability, expected efficacy plateau, or phase 2 data?

Response: Thank you for enquiry. The rationale for dose selection has been documented in Section 7.4.1 of the provided protocol ("Rationale for dose selection"). Briefly, the selection of ecnoglutide 1.2 mg was informed by findings from a phase 2 dose-finding study.¹⁰ In that study, the least-squares mean (standard error) change from baseline in HbA_{1c} at Week 20 was -2.39% (0.15) with ecnoglutide 1.2 mg, -1.90% (0.15) with 0.8 mg, -1.81% (0.15) with 0.4 mg, and -0.55% (0.15) with placebo. The HbA_{1c} reduction was superior in all ecnoglutide groups versus the placebo group and superior in the ecnoglutide 1.2 mg group compared with the ecnoglutide 0.4 mg group and the ecnoglutide 0.8 mg group. A consistent dose-response trend was also observed for fasting plasma glucose, 7-point SMBG, and bodyweight reduction. The safety and tolerability profiles of ecnoglutide 1.2 mg were also comparable to those of 0.8 and 0.4 mg. Therefore, 1.2 mg was selected as the primary maintenance dose for this study. According to the phase 2 findings, ecnoglutide 1.2 mg yielded a sufficient HbA_{1c} reduction and attainment of glycaemic control targets, suggesting that higher-dose evaluation is not required.

Furthermore, a quantitative pharmacology model using an E_{max} exposure-response framework predicted an HbA_{1c} reduction of -2.0% at 20 weeks for the 0.6 mg dose, representing approximately 77.7% of the maximum predicted effect (E_{max}) (unpublished data). Based on this modelling result, along with the expectation of improved tolerability (e.g., reduced gastrointestinal intolerance and decreased appetite) compared with the 1.2 mg dose, ecnoglutide 0.6 mg was also selected as a target dose.

Additionally, GLP-1-based therapies generally require higher doses for obesity than for T2DM, likely because the mechanisms driving HbA_{1c} reduction and weight loss differ. For example, semaglutide was evaluated at 0.5 mg and 1.0 mg in SUSTAIN-1 (T2DM) and at 2.4 mg in STEP-1 (obesity).^{11, 12}

2. The 1.2 mg group lost an average of 3.2 kg. Please briefly compare this with other GLP-1RAs and discuss its relevance for metabolic comorbidities.

Response: Thank you for the suggestion. In the manuscript, we have compared ecnoglutide with dulaglutide and supaglutide in terms of bodyweight reduction in Chinese patients from the same treatment setting and commented its potential cardiovascular benefits (Page 15, Lines 341 to 354). Briefly speaking, ecnoglutide appears to provide better bodyweight reduction effects than dulaglutide and supaglutide in Chinese patients. As there are no published phase 3 data for other approved GLP-1-based therapies for Chinese patients in the same treatment setting, we have not included comparisons with other GLP-1-based therapies.

3. Although the study includes a 28-week open-label extension following the 24-week double-blind period, the manuscript provides minimal information regarding its design and outcomes. While a brief mention of week 52 data is made in the Results section with reference to Supplementary Table 2 and Figure 2, the purpose of the open-label phase (e.g., assessment of long-term efficacy or safety) is not clearly defined in the Methods, and the week 52 data are not summarized in the main text. Please clarify the objectives, key endpoints, and analysis plan for the open-label period, and consider including a concise summary of week 52 findings in the Results section.

Response: Thank you for the enquiry. For the context, according to the *Technical Guidelines for Clinical Development of Drugs for Type 2 Diabetes in Adults* issued in 2023 by the Center for Drug Evaluation (CDE), National Medical Products Administration (NMPA), China,¹ to meet exposure and benefit-risk assessment requirements, pivotal clinical trials for non-insulin therapies in type 2 diabetes typically require at least two studies, usually: (1) a confirmatory trial evaluating the safety and efficacy of the drug as monotherapy, and (2) a confirmatory trial evaluating the benefits and risks of the drug in combination with metformin; placebo and an active control should be used as a comparator for the monotherapy study and the combination study, respectively.¹ Consistent with this recommendation, we have conducted two phase 3 trials (EECOH-1 and EECO-2) for the use of ecnoglutide in patients with type 2 diabetes. EECO-1, which is what this manuscript reports, evaluated the efficacy and safety of ecnoglutide monotherapy versus placebo in patients with type 2 diabetes inadequately controlled with diet and exercise alone or with a single oral hypoglycaemic agent. EECO-2 evaluated the efficacy and safety of ecnoglutide versus dulaglutide (another GLP-1 analogue) as add-on therapy in patients with type 2 diabetes inadequately controlled with metformin monotherapy.²

Meanwhile, according to the same technical guidelines,¹ when efficacy data from early-phase trials for the investigational drug are sufficiently robust, it is recommended that the placebo group not receive placebo treatment for more than six months due to ethical concerns. After that, patients in the placebo group may switch to the investigational drug. Safety data obtained during the period of investigational-drug treatment without a placebo control supplement the safety data collected during the placebo-controlled trial period. Following these recommendations, the double-blind period within our EECO-1 trial was set as 24 weeks, followed by an open-label extension period of 28 weeks to collect long-term efficacy and safety data of ecnoglutide.

The detailed results for the open-label extension period have been provided in **Supplementary Table 2** and briefly mentioned within the manuscript. Following your suggestion, we have expanded on the description of the changes from baseline in HbA_{1c} and bodyweight in each group (Page 12, Lines 253 to 261) but have not expanded on other outcomes due to the length restrictions, hoping this works for you.

4. HbA_{1c} reduction in the placebo group was -0.87% , which is higher than expected. Please discuss possible reasons, such as lifestyle changes, enhanced monitoring, or statistical regression. Fasting glucose and body weight also improved in the placebo group. Please consider whether this influenced effect size estimation and whether sensitivity analyses were performed.

Response: Thank you for the question. The strong placebo effect, which has been discussed within the manuscript (Page 16, Lines 377 to 390), may be due to the combined effects of several factors. Firstly, at every study follow-up, investigators provided participants with instructions on diet and exercise in accordance with the local clinical practice, and these instructions needed to be reinforced if participants had inappropriate habits (e.g., binge eating). These non-pharmacological interventions were expected to induce the HbA_{1c} reduction in the placebo group. Furthermore, in the placebo group, 67% (45/71) were treatment-naïve and the mean duration of type 2 diabetes was only 3 years. Therefore, these patients were expected to respond better to non-pharmacological interventions than those who previously received treatment. Additionally, a similarly strong placebo effect has been observed in a phase 3 trial of another GLP-1-based therapy

in Chinese patients from the same treatment setting.¹³

As specified by the statistical analysis plan, we have reported change from baseline data for HbA_{1c} at week 24 that were estimated using a mixed model for repeated measures (MMRM), with missing data imputed using the retrieved dropout-based multiple imputation approach. These statistical methods have been described within the manuscript (Page 19, Lines 488 to 498). We confirm that the results from the MMRM align with the descriptive data without imputation (**Table R1**), which indicates that statistical regression played no role in the higher-than-expected HbA_{1c} reduction in the placebo group. The MMRM and descriptive data have been placed below for your reference.

Table R1. Change from baseline in HbA_{1c} at week 24.

HbA _{1c} (%) at week 24	Ecnoglutide 1.2 mg (N=71)	Ecnoglutide 0.6 mg (N=69)	Placebo (N=71)
MMRM-estimated least-square mean change from baseline (95% CI)	-2.43 (-2.65 to -2.20)	-1.96 (-2.18 to -1.73)	-0.87 (-1.09 to -0.65)
Mean change from baseline (SD)	-2.40 (1.109)	-1.91 (1.060)	-0.88 (0.794)

The strong response in the placebo group narrowed the between-group differences between the ecnoglutide groups and the placebo group, and this does not increase the probability of a Type I error. Additionally, since the strong placebo effect was what was actually observed in this trial, there was no need to conduct a sensitivity analysis for this.

5. The 1.2 mg group had a mean increase of 5.4 bpm in heart rate. Although this is a known class effect, it may have implications for patients with underlying cardiovascular conditions. Additional data on QTc, nocturnal HR, or cardiovascular events would be helpful.

Response: Thank you for the request. This study did not collect nocturnal heart rate data, so we are unable to report them. We have added the most common adverse events of special interests ($\geq 5\%$ in any treatment group by preferred term) to **Table 3**. As for cardiac disorders (system organ class as defined in the Medical Dictionary for Regulatory Activities [MedDRA], version 27.0), the incidences at week 24 appeared to be numerically lower in the ecnoglutide groups than in the placebo group: 4.3% (3/69) in the ecnoglutide 0.6 mg group, 8.5% (6/71) in the ecnoglutide 1.2 mg group, and 12.7% (9/71) in the placebo group. Please note that these results are purely descriptive. The long-term impact of ecnoglutide treatment on cardiovascular conditions will be evaluated in future cardiovascular outcome trials. Furthermore, there are no obvious between-group differences in terms of the distributions of change from baseline in QTcF during the double-blind treatment period. The relevant data have been provided in **Table R2** below for your reference.

Table R2. QTcF during the double-blind treatment period.

Parameter	Ecnoglutide 1.2 mg (N=71)	Ecnoglutide 0.6 mg (N=69)	Placebo (N=71)
QTcF at baseline (ms)			
QTcF \leq 450	71 (100)	67 (97.1)	71 (100)
450<QTcF \leq 480	0	1 (1.4)	0
480<QTcF \leq 500	0	0	0
QTcF>500	0	1 (1.4)	0
Missing	0	0	0

Parameter	Ecnoglutide 1.2 mg (N=71)	Ecnoglutide 0.6 mg (N=69)	Placebo (N=71)
QTcF after baseline* (ms)			
QTcF≤450	67 (94.4)	68 (98.6)	66 (93.0)
450<QTcF≤480	2 (2.8)	1 (1.4)	5 (7.0)
480<QTcF≤500	1 (1.4)	0	0
QTcF>500	0	0	0
Missing	1 (1.4)	0	0
Change in QTcF from baseline* (ms)			
ΔQTcF≤30	63 (88.7)	64 (92.8)	64 (91.5)
30<ΔQTcF≤60	6 (8.5)	4 (5.8)	5 (7.0)
60<ΔQTcF≤90	1 (1.4)	1 (1.4)	1 (1.4)
ΔQTcF>90	0	0	0
Missing	1 (1.4)	0	0

*For each participant, QTcF after baseline and change in QTcF from baseline were calculated based on the highest QTcF value during the double-blind period.

References

- Center for Drug Evaluation, China. Technical Guidelines for Clinical Development of Drugs for Type 2 Diabetes in Adults. <https://www.cde.org.cn/main/news/viewInfoCommon/d5b2a1e8ee872ea1462a53a1da34a548>. (accessed 1 June 2025).
- He Y, *et al.* Efficacy and safety of cAMP-biased GLP-1 receptor agonist ecnoglutide versus dulaglutide in patients with type 2 diabetes and elevated glucose concentrations on metformin monotherapy (EECOH-2): a 52-week, multicentre, open-label, non-inferiority, randomised, phase 3 trial. *Lancet Diabetes Endocrinol* **13**, 863-873 (2025).
- Ji L, *et al.* Efficacy and safety of a biased GLP-1 receptor agonist ecnoglutide in adults with overweight or obesity: a multicentre, randomised, double-blind, placebo-controlled, phase 3 trial. *Lancet Diabetes Endocrinol* **13**, 777-789 (2025).
- Mu Y, *et al.* Efficacy and safety of once weekly semaglutide 2·4 mg for weight management in a predominantly east Asian population with overweight or obesity (STEP 7): a double-blind, multicentre, randomised controlled trial. *Lancet Diabetes Endocrinol* **12**, 184-195 (2024).
- Zhao L, *et al.* Tirzepatide for weight reduction in Chinese adults with obesity: The SURMOUNT-CN randomized clinical trial. *JAMA* **332**, 551-560 (2024).
- Tan TM. Is biased agonism helpful in the treatment of obesity with the GLP-1 receptor analogues? *Lancet Diabetes Endocrinol* **13**, 728-730 (2025).
- Scheen AJ. Ecnoglutide, a biased GLP-1 receptor agonist as a potential new player for type 2 diabetes management? *Lancet Diabetes Endocrinol* **13**, 818-820 (2025).

8. Guo W, *et al.* Discovery of ecnoglutide - A novel, long-acting, cAMP-biased glucagon-like peptide-1 (GLP-1) analog. *Mol Metab* **75**, 101762 (2023).
9. Shi LX, *et al.* Efficacy and safety of dulaglutide monotherapy compared with glimepiride in Chinese patients with type 2 diabetes: Post-hoc analyses of a randomized, double-blind, phase III study. *J Diabetes Investig* **11**, 142-150 (2020).
10. Zhu D, *et al.* Efficacy and safety of GLP-1 analog ecnoglutide in adults with type 2 diabetes: a randomized, double-blind, placebo-controlled phase 2 trial. *Nat Commun* **15**, 8408 (2024).
11. Sorli C, *et al.* Efficacy and safety of once-weekly semaglutide monotherapy versus placebo in patients with type 2 diabetes (SUSTAIN 1): a double-blind, randomised, placebo-controlled, parallel-group, multinational, multicentre phase 3a trial. *Lancet Diabetes Endocrinol* **5**, 251-260 (2017).
12. Wilding JPH, *et al.* Once-Weekly Semaglutide in Adults with Overweight or Obesity. *N Engl J Med* **384**, 989-1002 (2021).
13. Yan X, *et al.* Efficacy and safety of visepegenatide, a long-acting weekly GLP-1 receptor agonist as monotherapy in type 2 diabetes mellitus: a randomised, double-blind, parallel, placebo-controlled phase 3 trial. *Lancet Reg Health West Pac* **47**, 101101 (2024).

Comments from Reviewers

Reviewer #1 (Remarks to the Author):

There are no further comments.

Reviewer #2 (Remarks to the Author):

Reviewer #3 (Remarks to the Author):

When reviewing the manuscript I made 8 main comments that the authors should address. In the revised version of the manuscript which has been resubmitted to Nature Communications, the authors have considered all the comments and remarks except for point 6, but I agree with them for not accepting this comment as it is not in accordance with the journal requirements. I have no further comments to add.

Reviewer #4 (Remarks to the Author):

The authors responded appropriately to all the raised issues and comments. No further comments.

Response: We would like to thank everyone for the detailed feedback and for the second round of review. We are pleased that the revised version has adequately addressed all the raised comments.